# Family Caregiver Adaptation during the Transition to Adulthood of Individuals with Intellectual Disabilities: A Scoping Review

**DOI:** 10.3390/healthcare12010116

**Published:** 2024-01-03

**Authors:** Shivasangarey Kanthasamy, Nazleen Miskon, Joanna Barlas, Nigel V. Marsh

**Affiliations:** 1School of Social and Health Sciences, James Cook University, Singapore 387380, Singapore; shiva.kanthasamy@my.jcu.edu.au (S.K.); joanna.barlas@jcu.edu.au (J.B.); 2Department of Psychology, Institute of Mental Health, Singapore 539747, Singapore; nazleen_miskon@imh.com.sg

**Keywords:** neurodevelopmental disorder, intellectual disability, transition to adulthood, family adaptation, family caregiver, positive aspects of caregiving, quality of life

## Abstract

During the transition to adulthood, individuals diagnosed with intellectual disability (ID) and their family caregivers have unique experiences. This scoping review studies the sources of the family caregiver’s objective burden, support, coping mechanisms, positive caregiving, and quality of life to understand the caregiver’s adaptation process when the individual with ID transits to adulthood, according to Joanna Briggs Institute (JBI) Scoping Review methodology guidelines. The inclusion criteria included studies of family caregivers of any age who provide unpaid care and live with individuals diagnosed with ID who are transitioning to adulthood. Of 2875 articles identified, 12 published studies were included. The main themes included caregivers reporting dissatisfaction with the available adult services and exhaustion from being a caregiver. Overall, a vicious cycle of likely increased demands during the transition, with caregivers not being prepared to cope with these demands while concurrently being dissatisfied with the adult services system, leads the caregivers to develop a pervasive sense of helplessness. Future studies would benefit from recruiting caregivers from sources other than adult-only service centres and using qualitative (to identify the broad aspects of the key factors) and quantitative (to identify the significant differences between the key factors) methodologies.

## 1. Introduction

Caregiving for an individual with intellectual disability (ID) is a unique journey for each family caregiver. Due to the child’s cognitive limitations, the limited availability of social resources, possible changes in government policy, and caregivers’ ageing, the demands of caring for their son or daughter with ID who lives in a community setting may vary across the lifespan [1,2]. Family caregivers, the most prolonged and consistent care providers, continuously undergo adjustment and adaptation across their caregiving journey. This scoping review explicitly focuses on family caregiver adaptation in caring for individuals with ID transitioning from adolescence to young adulthood.

## 2. Individuals with ID Transitioning to Adulthood

An ID diagnosis implies an individual with significant impairment in cognitive and adaptive functioning. The diagnosis and severity level of the ID is determined by a clinician referring to criteria in manuals such as the American Psychiatric Association’s *Diagnostic and Statistical Manual of Mental Disorders* [3] and the World Health Organization’s *International Statistical Classification of Diseases and Related Health Problems* [4]. The assessment is based on the cognitive functioning scores and the clinician’s clinical judgment in assessing the individual’s ability to learn and adapt across conceptual, social, and practical skills in the early developmental (before 18 years old) period [5,6,7]. Based on a meta-analysis of 20 relevant articles, the estimated worldwide prevalence of ID ranges from 0.05 to 1.55% of the population [8]. Because of the aetiology of persistent neurodevelopment impairment, there is no “cure” for ID. In other words, ID is a permanent condition with life-long implications for management. The type and intensity of required support and training vary for each individual with the severity level of their ID (mild, moderate, severe, and profound). The severity level affects their ability for self-care, home living, health, safety, community living, and employment across the lifespan [9,10].

Individuals diagnosed with ID have significantly different challenges than neurotypical individuals during the transition to adulthood [11,12,13,14]. Generally, the transition from adolescence to young adulthood in neurotypical individuals is defined as progressing from supported environments such as school and family towards a more self-directed and independent life as an adult [15]. It is unreasonable to generalise and conceptualise the expectation of self-directed and independent living for individuals with ID, as their abilities and needs for support vary depending on the severity of their ID and their risk of becoming a member of a vulnerable population [16,17]. However, Blacher [18] highlighted that the transition for an individual with ID is expected to occur from 18 to 26 years old, specifically when the special education school services end and community-based adult services begin. 

Based on interviews with parents, Ferguson, Ferguson, and Jones [19] highlighted three different types of transition that coincide as the individual with ID turns 21: (a) bureaucratic transition, (b) family life transition, and (c) status transition. 

### 2.1. Bureaucratic Transition

Bureaucratic transition is the change that occurs between the professional and agency services that support the individual with ID and their family. In this case, the transition is from special education schooling to community-based adult services. Regarding the caregivers’ relationships with the professionals in adult services, four continuum types of patterns were categorised: abandonment, surrender, assimilation, and engagement. *Abandonment by professionals* is defined as when parents perceive that there is a lack of services provided by professionals, or they fail to find a suitable adult service placement. Ultimately, the parents feel that the responsibility of caring for their children lies entirely on them. *Surrender to professionals* is described as a process in which parents are conditioned to professionals’ opinions and explanations. Parents are perceived to be more prompt-dependent and might be more passive. *Assimilation with professionals* is defined as parents progressively adapting to provide the needed services for their child. For example, they might proactively set up home-based services, be more resourceful, and take on a more active or advocate role for their child during the transition. Lastly, *engagement with professionals* is a collaboration between the parents and professionals with separate functions but equal parental and professional responsibilities and engagement. These different types of relationships between the caregivers and professionals in adult services indicate the possible different styles of engagement and experiences during the bureaucratic transition and probably affect the outcomes for the individual with ID transitioning to adulthood. 

### 2.2. Family Life Transition

Family life transition describes the changes within the family system, such as daily routine and responsibilities (family role), when the son or daughter completes their special education program. Individuals with ID and their family members transit from six hours of structured engagement to new, unforeseeable adult services. The uncertainty of placement and the sustainability of adult services warrant parents’ effort and time. Hence, there is a need to accommodate the change in the caregivers’ routines and roles within the family system.

### 2.3. Status Transition

Status transition is defined as the change of status from child to adult. Like neurotypical individuals, the change to adult status in individuals with ID can be described by events such as becoming 21 years old, leaving the special education system, possibly acquiring and maintaining a job, and ultimately moving away from the family home to lead an independent life. However, caregivers of individuals diagnosed with ID have identified that the change in status was more about the intensity and frequency of supervision needed, rather than gaining total independence. The caregivers shared that they were apprehensive about the vulnerability risk and were ambivalent about letting go of their sons and daughters whilst yearning for independent adult children [19]. 

These three transition processes involving adolescents with ID occur concurrently, involving family caregivers, professionals, and adult services. Due to the impairment in cognitive and adaptive functioning of an individual with ID, identifying a key person in the support system will help aid the transition process in the community setting [20,21,22,23]. In the United States, most care providers are the families of children with ID [24]. Similarly, in Singapore, most individuals with ID live with, and are supported by, their family members in community settings. However, the family caregivers reported difficulty with their caregiver roles [25]. 

## 3. Family Caregiver Adaptation

Being a parent of a child born unexpectedly with a disability can be life-changing. Unlike other types of caregiving relationships (for example, spousal caregiving), family caregivers of adult children are considered distinct [26]. A scoping review was conducted to identify the factors influencing adult child caregivers’ well-being [27]. The caregiver caring for the adult child’s well-being was likely influenced by the caregiver’s role and the parent-child relationship quality. In other words, caregivers who have additional distinct roles other than caring for their adult children (for example, being sole income providers and providing care for their spouses) and poor relationships with the adult children for whom they are providing care are more likely to experience poor well-being [27].

Early family caregiver well-being research was based on the pathogenic paradigm of illness-focused behaviour [28]. The pathogenic paradigm emphasizes adverse outcomes, such as caregiver stress [29] and stigma [30] and their impact on caregivers’ mental health [31], physical health [32], and overall quality of life [33,34]. Similarly, in recent decades, various studies have focused on the family caregivers of individuals with ID based on the pathogenic paradigm [35,36,37]. The psychological interventions suggested based on the outcomes of these studies are equally geared towards reducing adverse consequences [38]. However, current researchers are taking steps to promote health by understanding the journey of caregiving based on a health paradigm rather than emphasizing the negative aspects of caregiver stress [39,40], while still acknowledging the family caregivers’ struggles. The health paradigm focuses on actualizing the capacity of the caregivers and increasing the caregivers’ health and well-being [28] by focusing on themes associated with personal growth and rewards for caregivers in raising children with developmental disorders [40].

Although much research is available about individuals with ID and their family caregivers’ well-being, there are still gaps in understanding the “successful transition” [41]. Thus, understanding caregiver adaptation by mapping the factors that trigger and regulate the adaptation process will aid in understanding both the positive and negative factors of the caregiving journey, specifically during the transition to adulthood of an individual with ID.

## 4. Review Questions

A preliminary search in MEDLINE, the Cochrane Database of Systematic Reviews, and JBI Evidence Synthesis was conducted in February 2022 and found no systematic review, scoping review, or protocol related to this topic. The objective of this scoping review was to explore the breadth or extent of the available research literature and to map and summarise the evidence from the literature on family caregiver adaptation, specifically during the transition to adulthood, when caring for an individual with ID. Sources of the caregiver’s objective burden, support, coping mechanisms, positive caregiving practices, and quality of life were reviewed to understand the caregiver’s adaptation process when the individual with ID transits to adulthood. In addition, the review also examined the methodology of the research conducted on this topic.

Based on the available literature, the following questions were generated:What are the reported sources of the family caregivers’ objective burdens (stress related to the individuals with ID and caregiver factors) during the transition to adulthood of individuals with ID, for each severity level (mild, moderate, severe, and profound)?What kinds of support and resources do the family caregivers report they need to manage individuals with ID, for each severity level (mild, moderate, severe, and profound), during the transition?What coping mechanisms do family caregivers commonly use during the transition to adulthood of individuals with ID?What positive aspects of caregiving help family caregivers during the transition to adulthood of individuals with ID?What is the family caregivers’ quality of life like (i.e., physical and mental health, perceived subjective burden) during the transition to adulthood of individuals with ID for each severity level (mild, moderate, severe, and profound)?Are other factors related to caregiving, such as the caregivers’ anxiety levels or expectations about the future independent living of the individuals with ID, influencing the caregivers’ quality of life during the transition?

## 5. Method

This scoping review was conducted according to the Joanna Briggs Institute (JBI) Scoping Review recommended methodology guidelines and was based on a three-step search strategy [42]. The JBI framework of Population, Concept, Context (PCC) was followed.

### 5.1. Inclusion Criteria

#### 5.1.1. Population

Family caregivers are the caregivers who live with and provide unpaid care to an individual diagnosed with ID. Therefore, literature on collective families, siblings, teachers, and paid or professional caregivers was excluded. In terms of the diagnosis, the primary diagnosis was ID, which includes genetic disorders such as Down syndrome and Fragile X syndrome. Other specific neurodevelopment disorders such as autism spectrum disorder (ASD) or Asperger’s syndrome, attention-deficit/hyperactive disorder (ADHD), specific learning disability (SLD), conduct disorders, vision and hearing impairments, dementia, as well as other mental illness, were excluded.

#### 5.1.2. Concepts

The concepts of this scoping review were family caregiver adaptation and individuals with ID transitioning to adulthood. This scoping review included literature on factors related to the adaptation of family caregivers when managing individuals with ID [43]. The Transactional Theory of Stress and Coping [44] defines perceived stress as the outcome of a cognitive imbalance between the interpretation of the stressors and the available resources that an individual considers themself to have to cope with the stress. The concept of “adaptation” was broadly defined, as there was an objective to capture as much literature as possible. Thus, the factors on “adaptation” included, but were not limited to, family caregivers’ objective and perceived burden, support and resources, coping mechanisms, positive caregiving, and quality of life. Therefore, any factor related to the family caregivers’ quality of life during the transition was included.

Another concept was that of the individual with ID transitioning to adulthood. The definition of adulthood from the General Social Survey highlighted being financially independent individuals, completing school, leaving home, and working full-time [45]. However, for individuals with ID, the expectation of achieving independent living is only sometimes realistic, depending on the range of severity from mild to profound. In this scoping review, the transition to adulthood was conceptualised as completing formal schooling (i.e., finishing high/secondary school). To broaden the search, “transition to adulthood” was included to capture literature on individuals with ID who did not attend or graduate from school. Literature on individuals with ID still attending school, transitioning from primary to secondary school, or transitioning from paediatric medical service was excluded.

#### 5.1.3. Context

There was no restriction on context (cultural, geographic location, specific race or gender, specific settings) for this scoping review. All research literature meeting the inclusion and exclusion criteria for the participant and concept were included in this scoping review.

### 5.2. Types of Evidence Sources

This scoping review included academic literature sources from journal articles, conference papers, dissertations, and book chapters. The sources included but were not limited to study designs such as analytical observational studies, prospective and retrospective cohort studies, cross-sectional studies, and single or multiple case studies. Non-academic literature such as, but not limited to, newspaper articles, personal blogs or reviews, theoretical and opinion papers, and sources lacking original research were excluded from this scoping review.

### 5.3. Search Strategy

The search extended to published and unpublished (grey or tough-to-locate literature) sources. Only literature published in English was included since no resource was available to translate literature not written in English. There was no date range limitation in this scoping review.

#### 5.3.1. Published Source Search

**Initial Limited Search.** An initial limited search in databases such as APA PsycInfo and Scopus was performed to identify literature on this topic. Guidance was received from a librarian. Google Search was used to find synonyms for “transition”, and relevant terms were selected; transit, change, changing, move, or moving. The initial search terms that were used were ((school OR adult*) AND (transit* OR chang* OR move OR moving)) AND (parent* OR caregiver* OR father* OR mother*) AND (“intellectual disability” OR “intellectual disabilities” OR “intellectual disorder”). The key sources were screened for keywords and index terms.

**Keywords and Index Terms**. The keywords in the titles and abstracts of relevant literature (Appendix A) and the index terms were analysed to develop a full search strategy. After an iterative process, the identified keywords and index terms were adapted for all included databases (ProQuest (APA PsycInfo and ERIC), Scopus, and Web of Science; Appendix B). The search was completed in July 2023, and 2558 citations were identified for this review.

**Reference List of Identified Sources.** The reference list of all included literature (after the full-text screening) was assessed. There was no additional literature identified via the reference list.

#### 5.3.2. Unpublished Source Search

Unpublished studies of grey literature were searched for using the following sources; ProQuest Theses and Dissertations (PQDT; https://www.proquest.com (accessed on 13 July 2023).) and Google Scholar. The search terms used were ((school OR adult*) AND (transit* OR chang* OR move OR moving)) AND (parent* OR caregiv* OR father* OR mother* OR care giv*) AND (“intellectual disability” OR “intellectual disabilities” OR “intellectual disorder” OR “mental retardation”). A total of 317 sources were identified.

#### 5.3.3. Source of Evidence Selection

All the citations identified from the above search strategy were imported into data management software, *EndNote 20* (Clarivate Analytics, PA, USA). A total of 999 citations were removed automatically using *EndNote 20* before screening. The process of citation selection was divided into two phases: title and abstract screening and full-text screening.

**Title and Abstract Screening.** During the screening process, 1876 citations were uploaded from *EndNote20* to *Abstrackr* (http://abstrackr.cebm.brown.du (accessed on 27 July 2023)). The titles and abstracts were screened based on guidelines outlined by Polanin et al. [46] to identify appropriate sources effectively while minimising bias. A screening tool (Appendix C) was drafted based on Polanin et al.’s [46] guidelines. Two screeners, the first and second authors, screened the titles and abstracts. The screeners learned and pilot-tested the screening tool by screening the abstracts using 30 abstracts. Once consensus was achieved with the screening tool, the screeners proceeded to screen the titles and abstracts independently. Each abstract received independent double screening. The screening was monitored continuously, and the pair agreement was calculated. Polanin et al. [46] suggested maintaining the agreement within 75%. High screener disagreement rates (less than 75% agreement) may indicate issues related to the screening tool or the training received. Screeners met and reconciled after completing 20% to 30% of the abstracts. Overall, the agreement achieved for this scoping review was 76%. The agreed-upon process for reconciling disagreements throughout the abstract screening process was followed. Any further disagreements were discussed with a third reviewer to obtain a consensus. A total of 252 citations were identified based on the screening tool inclusion and exclusion criteria for full-text screening.

**Full-Text Screening.** The two screeners conducted a full-text literature screening, following the title and abstract screening process. After reading the full text, the screeners further assessed whether or not the papers met the inclusion criteria. Literature that did not fit the inclusion criteria was excluded. Authors of the screened citations were contacted when more information or clarification was needed. The details of both the included and excluded papers and brief explanations of reasons why papers were excluded are presented via a Preferred Reporting Items for Systematic Reviews and Meta-analyses (PRISMA) [47] flow diagram (Figure 1). A total of 12 studies were identified based on the full-text screening tool’s inclusion and exclusion criteria to be included in this scoping review (Appendix D).

## 6. Results

A total of 2875 citations were identified from published and unpublished source searches. After removing duplications and screening based on the inclusion and exclusion criteria, 12 full-text studies were included in this scoping review.

### 6.1. Characteristics of the Studies

The characteristics of these 12 studies are summarised in Table 1. The year of publication of these studies ranged from 2008 to 2022, with the highest number of studies published in 2020 (*n* = 3). The studies were conducted in the United Kingdom/England (*n* = 3), Australia (*n* = 3), Canada (*n* = 2), and one study each in Ireland, Israel, Norway, and South Africa. Eleven of the studies were qualitative, and one was quantitative. All qualitative studies used semi-structured one-to-one interviews, and one included a focus group. Ten studies were cross-sectional, and two were longitudinal in design.

In terms of participants, the number of family caregivers ranged from 2 to 301 caregivers, primarily females. None of the studies included individuals with ID as participants, but most reported the children’s ID, age range, and severity level.

### 6.2. Review Findings

#### 6.2.1. Question 1

What are the reported sources of the family caregivers’ objective burden (stress related to individuals with ID and caregiver factors) during the transition to adulthood of an individual with ID, for each severity level (mild, moderate, severe, and profound)?

**Individuals with ID.** According to Rapanaro, Bartu, and Lee’s study [48], 94 caregivers of 119 respondents reported having at least one stressful event in the past 12 months. The events were categorised into five categories: (a) young adult behaviour/conduct (38.3%), (b) issues with the service provider (22.3%; refer to *Objective 2*), (c) independence issues (16%), (d) health problem (13.8%), and (e) young adult vulnerability (9.6%).

Specifically, on *young adult behaviour/conduct* (38.3%), the behaviours were described as being aggressive, socially inappropriate (stealing and disruptive at work) and sexual conduct (having unprotected sex). Some caregivers identified the increase in challenging behaviours as being due to the difficulty of adjusting to adult programs [48]. Gauthier-Boudreault, Gallagher, and Couture [52] added that due to the lack of adult services, caregivers of individuals with ID in the profound severity range reported that it affected their children by reducing their children’s capabilities and abilities, and increasing weight gain, boredom, and challenging behaviours.

“*Since leaving school, life has not been the same for my son. He loved school and was extremely happy. We found work for him [in a sheltered workshop] which he wanted to attend… He has changed positions many times to try and make him happy…Due to the unhappiness and frustrations he copes with at work, in a sheltered workshop, he has become very aggressive…*”(Rapanaro et al., p. 37 [48])

The transition to adulthood for an individual with ID also indicates the incongruence between their physical and cognitive–emotional development. Hubert [50] highlighted the challenges the mothers of children with ID in the severe to profound range undergo. As a child grows, their physical size and strength are bigger and stronger than their ageing mother’s. It is more challenging to contain incontinence, maintain appropriate hygiene levels during menstruation, and manage violent and overt sexual behaviours during the transition. Similarly, caregivers of individuals with ID in the severe range in Biswas, Tickle, Golijani-Moghaddam, and Almack’s study [53] expressed that their stress was triggered when they were unsure how to support the child’s sexual development. Nucifora, Walker, and Eivers [59] reported that caregivers felt distressed when they were aware of the gaps between their child’s capacity and an individual of similar age with no ID (neurotypical).

“*When you see the gap gets bigger, every year she gets older, more things drop off that she is not able to do.*”(Nucifora et al., p. 5 [59])

Regarding *independence issues* (16%), the caregivers reported concerns about their child’s lack of independence or seeking of independence. Lack of independence was reported when the child could not travel using public transport and had difficulty dealing with legal and financial issues independently. Seeking independence was described as the child’s desire for freedom away from the family home but having difficulty coping with social demands. Caregivers can no longer guide the child due to the child’s legal age and need legal authority such as power of attorney [48]. Caregivers shared that ongoing involvement was needed even when their contact with the child had reduced in cases when the child had moved out of the family home [51]. Caregivers reported extra responsibility in teaching their child skills needed to be independent such as travelling in public buses [59].

*Health problems* (13.8%) were reported to involve caregivers and their child’s health. Some caregivers reported that their child had a comorbid physical disability. The associated demands, such as negotiating treatment and care plans as well as witnessing their child’s pain and discomfort, were reported to be stressful by the caregivers. Due to caregivers’ ageing, arranging respite care and funding was difficult, and their child reported being upset due to the changes [48].

Lastly, *young adult vulnerability* (9.6%) relates to abuse/harassment and being negatively influenced by others. Caregivers expressed concern that their children might be taken advantage of [48]. Biswas et al. [53] shared similar findings that caregivers’ negative thoughts about adult services and institutions trigger their fear of their child’s safety due to the risk of being vulnerable in the community. In addition, caregivers are more worried if the child has limited verbal ability and cannot report the possible abuse. These caregivers view barriers to adulthood as their inability to plan for adulthood-related activities, to count on professional support, and the child’s limited cognitive and social skills as well as lack of personal responsibility.

**Caregivers.** Regarding caregivers’ employment, Roos and Sondenaan [56] highlighted that some caregivers of individuals with ID in the profound range cannot sustain their employment because they need to take care of their children. Another study conducted by Gur, Amsalem, and Rimmerman [55] highlighted in detail that almost half of the caregivers of an individual with all ranges of ID were employed, and about 47.5% of the caregivers were out of the workforce. Among those who were out of the workforce, 44.1% of the caregivers reported that they stopped due to the need to care for the child with ID, but the ID range was not specified in this study.

Gur et al.’s study [55] also outlined that apart from employment, a significant difference was found in caregivers of individuals under the age of 21 who can spend more time outside their home compared to the groups of caregivers of individuals with ID above 21 years of age. However, no significant differences were found among the caregivers of individuals with ID aged 21 to 30 and above 31 years old in the number of hours spent outside their home. Overall, almost half of the caregivers (41.2%), reported spending more than 15 h a day caring for their child with ID, and the time spent was higher when the caregivers cared for individuals with ID who were above 21 years old. Nucifora et al. [59] added that caregivers reported reducing their own “independence” due to the caregivers’ strong sense of responsibility related to caregiving needs during the transition period.

“*The fact that it sort of restricts our independence in a way because we’re having to supervise them.*”

“*I also don’t have any free time um, it’s a go from 5 am through to 10 pm at night.*”(Nucifora et al., p. 8 [59])

The caregivers reported that the financial burden significantly affected some of the families. Beyond that, “*negotiating everyday occupation*” was a struggle as caregivers needed to make practical household arrangements with different routines during the transition period [57]. Nucifora et al. [59] highlighted that caregivers have an ongoing sense of responsibility and the need to supervise and support their child diagnosed with ID, so the caregivers develop a fear of ageing. The caregivers were concerned about how their child would be supported when the caregivers could not do so due to the caregivers’ ageing factor.

#### 6.2.2. Question 2

What kind of support and resources do the family caregivers report they need to manage individuals with ID for each severity level (mild, moderate, severe, and profound) during the transition?

**Formal Support**. Gillan and Coughlan [49] highlighted a list of barriers to the transition process shared by caregivers managing individuals with ID of mild severity. Examples included a lack of information provided, a lack of accommodation to individual needs for vocational service, a lack of coordination between child and adult services, a lack of parental involvement, and a lack of “real” alternatives for a vocational training provider. Other barriers included being on a waiting list for the services, staff not adequately communicating, and staff “not listening” to caregivers. Beyond the adult service provider, caregivers reported negative experiences related to the environment and training in the employment service setting. The caregivers expressed the need for adequate formal support due to their children’s needs. 

Isaacson, Cocks, and Netto [51] reported that the caregivers had difficulties accessing formal support to manage their children with Down syndrome with moderate to high needs, such as a lack of information provided and long-term frustration with the system (filling out many forms). However, both families were ultimately satisfied with the trainers assigned to the children. 

Biswas et al. [53] reported that caregivers of individuals with ID in the severe range reported perceived barriers to formal support and that little information was provided related to the transition. The professionals’ understanding of the “normative” transition to adulthood contradicted caregivers’ and their children’s need for support. In other words, the professionals shared with the caregivers that since the child was above 18 years old, they were not the caregivers’ “responsibility anymore”. Thus, more negotiation between the parents and professionals regarding the management plan is needed. The current policy that promoted autonomy and independence evoked different views among the caregivers. Some caregivers in this study shared that it was helpful and advocated for “normal” adult development. On the other hand, some caregivers were concerned about the possible risk involved when their child is encouraged to pursue “normal” adult development due to the possible risk of being a member of a vulnerable population. Caregivers also reported that they required professional support for psychosexual development.

One of the themes that emerged for caregivers managing individuals with ID in the severe range was the need to manage the transition on their own; the “burden was theirs” [57]. Even though the caregivers expected some support, they did not initiate or seek opportunities to engage with the support system. They handled the situation as it arose on their own. Some caregivers reported that the presence of a non-governmental organisation (NGO) that offered to recruit their children was helpful with the transition. 

Roos and Sondenaan [56] highlighted that the caregivers of individuals with ID of profound severity reported concern about the waiting list to move out of the family homes. The caregivers needed to know how long they had to wait and where the child would be moving. The caregivers reported dissatisfaction with the details given on the services in the adult home and the absence of respite care during the waiting period. Once the child was placed, the caregivers reported concerns about the lack of fit for their children’s needs with the co-residents, such as age, functioning level, and interest. Furthermore, the caregivers must visit their children frequently to help with their practical and leisure needs. Compared to the service received in the children’s homes, caregivers also expressed concern about the reduced services in the adult homes. For example, caregivers did not receive updates from the adult home about the child’s well-being. On the other hand, one out of nine caregivers expressed satisfaction with the placement. One caregiver reported that although it was concerning for the child to move out to the family home, it was perceived to be beneficial for the child to receive care from other staff and residents besides the “exhausted” parents.

Gauthier-Boudreault et al. [52] defined the themes of the caregivers’ need to manage individuals with ID of profound severity. The caregivers expect material, informational, affective, and cognitive support. *Material support* was defined as the need for services and resources to solve the practical needs of everyday life. Caregivers reported possible barriers that they faced to receive material support. For example, late transition planning, lack of collaboration between organisations, professionals not being aware of their roles during the transition, and shifting of their responsibility to other areas caused service gaps in areas such as day activity centres and respite care, and difficulty accessing adult health services. *Informative support* was the need for support related to difficulty accessing information during the transition phase. *Affective support* was the lack of support for the caregivers to share and socialise. *Cognitive support* was related to the lack of intervention catered to their child with profound ID in the day activity centre and lack of expertise in the adult health care professionals. 

There were reported stressful “service provider issues” events described by the caregivers of individuals with ID of all severity ranges, which accounted for 22.3% of those who had at least one stressful event in the past 12 months [48]. The perceived difficulties were related to issues accessing services and dissatisfaction with the organisations or the service staff. Examples of the concerns were related to the difficulty in gaining and maintaining adult services due to money and support factors, the application for accommodation service not being approved, the service providers mistreating the caregivers or the child with ID (lack of support and assistance), and the work situation not being satisfactory. Caregivers reported that they needed to advocate on behalf of their children. 

“*[her] work situation hasn’t to date been satisfactory. The work is not challenging enough and my child eventually wants to quit, which causes anxiety all round. This time it eventually caused her to have a re-bout of depression, which needed counselling….*”(Rapanaro et al., p. 37 [48])

Wilcox, McQuay, and Jones [54] highlighted in their case studies that the mothers of individuals with ID in the mild to severe ranges reported that they needed support for a successful transition in the form of adult service providers. They expected better communication and follow-up about the transition process, expressed a need for more information on the available training and resources, and wanted a more planful transition. One mother highlighted the system strain as the service workers were overworked, and the staff turnover was high. The mothers also shared that a poor placement fit for the adult service, the paperwork, and the burden of advocating for the child had been stressful.

“*I think PDD workers are very over-worked.*”

“*The turnover is great. We went through so many caseworkers and they’re so overloaded … they’ve got so many things on their plate*”(Wilcox et al., p. 11 [54])

Gur et al. [55] reported no significant differences in caregivers’ social participation as a function of ID severity. Similarly, no significant differences existed among the caregiver groups’ per capita income and out-of-pocket expenses. However, there were substantial differences in the type of support received among the caregivers’ groups. Caregivers of persons under the age of 21 received significantly more support from NGOs and used more services than caregivers of persons over 31. There were no significant differences in the type of support received among the caregivers of persons aged 21 to 30 and above 31 years old.

Codd and Hewitt [58] highlighted that caregivers reported negative experiences with adult service providers, irrespective of ID severity. Some positive experiences were reported, but they needed to be more consistent. Caregivers reported a lack of support and trust and perceived professionals to be incompetent in the adult compared to the child services.

Nucifora et al. [59] reported that the increase in funding and flexibility in service selection via the personalised support service in Australia, known as the National Disability Insurance Scheme (NDIS; established in 2016), does help to increase some of the caregivers’ levels of trust towards the support staff. Although the NDIS was perceived to be more flexible than the previous services, caregivers raised concerns that the other government-based services were not flexible enough. The shift from school to new adult services was perceived as lacking structure and not helping the caregiver transition. In addition, caregivers reported that the legal definition of “adulthood” complicates access to certain services.

**Informal Support**. Isaacson et al. [51] reported that the caregivers of moderate to high-need people with Down syndrome were supported by their extended family members and friends when needed. Biswas et al. [53] highlighted that those caregivers of individuals with ID in the severe range reported perceived support from informal sources such as a supportive network. Caregivers of persons under 21 years of age received significantly more support from family and friends and used more services than caregivers of people over 31 [55]. Nucifora et al. [59] highlighted that some caregivers shared that having a supportive family does help with the caregiving demands during the transition to adulthood, as the caregiving demands are being shared. Having supportive friends was seen as necessary support for both caregivers and their children with ID.

“*When I see Daniel with his mates … I just feel confident that they’ll be there for life… that’s when I feel most relaxed*”(Nucifora et al., p. 8 [59])

However, some caregivers, especially the father in this study, reported feeling reluctant to share their concerns to avoid being burdensome to others [53]. The mothers caring for individuals with ID in the severe to profound range shared negative experiences of being socially isolated as their extended family members became harsher and fearful of their adult child [50]. 

#### 6.2.3. Question 3

What coping mechanisms do family caregivers commonly use during the transition to adulthood of individuals with ID?

**Information Gaining.** There were different types of coping mechanisms reported in each study. One common theme was the need to gain information by communicating with others and taking a proactive role. The caregivers of individuals with ID in the mild range reported that regular communication with the adult service providers helped them cope with the service system [49]. Biswas et al. [53] reported that caregivers of individuals with ID in the severe range coped by researching to gain more information about the transition process generally. Caregivers of individuals with ID in the profound range reported that they preferred to be connected with other families to plan the housing option independently without depending much on the service providers [56].

**Accessing the Support System and Proactive Roles**. Caregivers reported that “accessing resources and a support system” such as “faith in a higher power, families, personal skills and abilities, organisations and influential people in the community” helped them deal with the transition when managing individuals with ID in the severe range [57]. Six of eight families said they are always in “fighting” or “battling” mode to receive the appropriate support, resources, and employment for their adult child. In addition, caregivers reported taking a proactive role in their children’s management [49]. Similarly, caregivers shared that they had to access support systems such as social connections and government-based services to cope with the transition. Additionally, caregivers had to constantly advocate with the appropriate government-based services for their children [59].

“*You’ve got to advocate the whole time within the healthcare system to get people to look at her as an individual…. A continual battle*”(Nucifora et al., p. 6 [59])

**Setting up Routines for Caregivers.** Ellman, Sonday, and Buchanan [57] reported that caregivers of individuals with ID in the severe range developed coping mechanisms as they struggled with the changes during the transition. One of the coping mechanisms was “*setting up a routine*”. Some caregivers volunteered at the NGO while their children attended the activities. It helped them to keep an eye on their child. Some caregivers reported that the NGOs were short of staff, so they decided to help. At the same time, others reported that they volunteered to secure their children’s placement at the organisation.

**Managing Expectations.** Caregivers’ managing and accepting reasonable expectations of their children was reported to be one of the coping mechanisms. Caregivers of individuals with ID in the severe range who do not perceive the transition to adulthood to be based on chronological age may have lower expectations and be more accepting of the change in the “learning disabilities sub-culture”. Lower expectations lead these caregivers to have a more positive outlook on the children’s adult life and to be able to manage their worries [53].

#### 6.2.4. Question 4

What positive aspects of caregiving help family caregivers during the transition to adulthood of individuals with ID?

**Mixed Feelings.** Rapanaro et al. [48] reported that almost half of the caregivers of individuals with all severity ranges of ID (43 out of 94 caregivers, 45.7%) reported positive outcomes of stressful episodes and almost three-quarters of the caregivers reported overall perceived benefits from the chronic demands of being a caregiver (77 out of 119 caregivers, 64.7%), for only positive or both positive and negative aspects. Similar to Rapanaro et al. [48], Ellman et al. [57] reported that caregivers of an individual with ID in the severe range reported “*mixed feelings*”. The situation factor, such as the physical and social environment, influenced the fluctuation of positive and negative experiences. It was reported that there were more negative than positive experiences.

**Sense of Fulfilment and Pride.** Rapanaro et al. [48] drew attention to the theme of fulfilment and pride (a sense of purpose and fulfilment), personal growth (tolerance, patience, appreciation of life, and greater acceptance), enhanced social networks, and absence of specific care demands (no need to dress, bathe, or entertain as the child gains independence in their daily living activities). Hubert [50] also reported a related theme of fulfilment and pride. Although the mothers caring for individuals with ID in the severe to profound range in the study were socially isolated, they reported that having positive relationships with their children was rewarding. They felt proud of their children and themselves through their caregiving journey.

**Gratitude.** Gillan and Coughlan [49] reported that all 12 caregivers of individuals with mild ID reported positive experiences at the beginning of the transition. One positive experience was that at least one service provider provided the needed support in one way or another. Thus, although there was a concern about inadequate services, as reported above, the caregivers expressed gratitude for having at least one service provider to guide them.

**Benefits for Caregivers**. There were two categories of themes: (a) enhanced caregivers’ resources and growth, and (b) seeking more formal and informal support [48]. For example, caregivers reported that these stressful events had helped them to be more assertive and determined to have an increased understanding of their needs for the transition. The caregivers also shared that they managed to get new and improved support from the community and possibly became more close-knit with family.

**Benefits for Individuals with ID**. Similarly, for their children with ID, caregivers reported that their children could learn new coping skills, improve challenging behaviour, and increase confidence and maturity [48]. Gillan and Coughlan [49] also reported that the children’s capability for positive adjustment was rewarding for the caregivers during the transition. Seven out of twelve caregivers reported that their child was observed to have increased in confidence. Ten out of twelve caregivers claimed that their child enjoyed their work or training setting.

#### 6.2.5. Question 5

What is the family caregivers’ quality of life like (i.e., physical and mental health, perceived subjective burden) during the transition to adulthood of individuals with ID for each severity level (mild, moderate, severe, and profound)?

**Negative Emotions Due to Lack of Support**. Overall, the caregivers of individuals with mild ID reported a consistent theme of perceived stress and anxiety due to the nature of the service system. Nine out of twelve caregivers reported frustration “dealing with the inflexible and unresponsive service,” and four reported helplessness due to the lack of substitute services [49].

Caregivers of individuals with Down syndrome who had moderate to high support needs expressed that they were physically and mentally tired. One of the caregivers reported having “mild depression during the initial period of moving out”. However, after the children left the family home, the caregivers could spend more time with their spouses [51].

Similar to this, Hubert [50] reported that caregivers caring for individuals with severe to profound ID were “emotionally and physically exhausted” as the daily needs to care for their children were time-consuming and exhausting.

Caregivers of individuals with severe ID reported “feeling uncertain and confused” due to unfamiliarity with the transition period. Decision-making was difficult and their daily life activities felt restricted during the transition period due to the lack of information received during the pre-transition period. There was much uncertainty and not knowing what to expect. A sense of loneliness was reported as the caregivers would not approach their family and friends for help and vice versa during the transition [57].

In Norway, an application for an apartment is made once the child reaches 16 years of age. However, the child’s placement usually will be based on a crisis [56]. Specifically, caregivers of individuals with profound ID expressed an “unsustainable burden of care” while waiting for housing, mainly after their children were 18 years old.

“*I don’t think they (staff in the municipality) understand and see signs of an exhausted body—they understand it’s tough, but no one realises how tough it really is—I have parents who have helped for many years—but they are also starting to get old—I had to call them one night when I couldn’t do it anymore—there was a crisis—then they came. It hits a whole family.*”(Roos et al., p. 5 [56])

Irrespective of the severity range, caregivers of individuals with ID reported feeling sad and tired of being the “fighters and advocates” for their children. The parenting chores may change, but the intensity and effort needed during the transition are still the same [58]. Gur et al. [55] reported that the average score on the well-being measures of caregivers of people with ID aged 31 years and above was significantly lower than that of persons caring for people with ID aged 21 to 30. Caregivers of persons aged 31 years and above were reported to be more frustrated than caregivers of persons under 21. The average score on the life satisfaction measure of the caregivers caring for children with ID was the lowest when the child was above 31 years old, followed by the caregivers of children below 21. Caregivers caring for children with ID aged 21 to 30 and above 31 years old reported being sadder than caregivers caring for a child under 21.

Rapanaro et al. [48] highlighted that caregivers of individuals of all ranges of ID had common themes revolving around negative feelings such as guilt, resentment, anger, fear of repeat events and the future, depression, hopelessness, low self-worth, mistrust, and wishing that they did not have a child with a disability. Caregivers reported extra resource demands, financial strain, and failure to access adult services. The caregivers further highlighted the negative impact on family relationships, where the other siblings were affected (due to fear or embarrassment), marriages were strained, and the relationships with their children with ID diagnoses were affected due to the need for caregiving duties to be performed. Caregivers also reported a loss of freedom and independence in their personal aspirations and social activities.

“*it has been bloody stressful—so much so, I nearly died of an asthma attack. I was unconscious, [my] husband resuscitated me…lifting has caused lower back pain…and occasional feelings of impending heart attack due to[the] physical exertion of lifting and carrying*”(Rapanaro et al., p. 42 [48])

#### 6.2.6. Question 6

Are other factors related to caregiving, such as the caregivers’ anxiety levels or expectations about the future independent living of the individuals with ID, influencing the caregivers’ quality of life during the transition?

**Independence and Vulnerability Risk.** Gillan and Coughlan [49] emphasised that caregivers of individuals with mild ID shared their hesitation to provide more independence to their adult children. The caregivers’ hesitation was due to their apprehension of their adult child being vulnerable because of cognitive limitations. Six and nine caregivers reported concerns about others taking advantage of their adult children and the children lacking assertive skills in work settings, respectively. Consistently, caregivers of individuals with moderate to profound ID shared their concern about their children’s independence, considering the vulnerability risk [51]. Similarly, Codd and Hewitt [58] highlighted that irrespective of severity level, caregivers of individuals with ID reported their dilemma of “*letting go and separation*” and apprehensiveness to support independence due to their children’s vulnerability and their lack of trust in adult services. Nucifora et al. [59] added that the caregivers struggled cognitively and emotionally with the degree of independence and caregiver monitoring needed during the transition to adulthood. The caregivers were aware of their children’s vulnerability and, due to fear of abuse, the caregivers reported difficulty trusting the support providers.

“*With James, it is still like having a kid in some ways because he still needs the same amount of care*”(Isaacson et al., p. 275 [51])

“*It’s hard to let go of doing it for so long. It’s just purely your instincts and your protection*”(Nucifora et al., p. 7 [59])

**Caregivers’ Anxiety.** Codd and Hewitt [58] highlighted that irrespective of severity level, caregivers of individuals with ID reported uncertainty and worries regarding the sudden change to unknown adult services. Hubert [50] emphasised that the caregivers of individuals with severe to profound ID reported being overprotective and anxious due to the need to manage high-needs children with a perceived lack of “adequate” support. For all severity levels, caregivers of individuals with ID aged above 31 years were more concerned about their children’s diagnosis than caregivers of individuals with ID aged 21 to 30 years [55].

## 7. Discussion

This scoping review aimed to study the sources of the caregiver’s objective burden, support, coping mechanisms, positive caregiving, and quality of life to understand the caregiver’s adaptation process when the individual with ID transits to adulthood. The caregiver’s objective burden was divided based on the needs of the individual with ID and the needs of the caregiver. Regarding the caregiving demands related to individuals with ID, stressful events were divided into young adult behaviour/conduct [48,52] and incongruency between the physical and cognitive–emotional development of individuals with ID [59]. In other words, caring for an individual with ID who is physically growing stronger and bigger with challenging behaviours [50] and psychosexual development needs [53] was reported to be stressful. The prevalence of challenging behaviour in a study conducted with 265 adults with ID was 18.1% (about one-fifth of the adult population with ID, [60]). The challenging behaviours in adults with ID are more persistent and stable over time, whereas there is a tendency for the challenging behaviours in typically developing children to decrease over time [61]. No studies have been conducted yet that make comparisons between the prevalence of challenging behaviour in individuals with ID pre-, during, and post-transition to adulthood. Three studies discussed the caregiver’s objective burden related to independence issues (individuals with ID either had a lack of independence skills or were seeking independence) and the need for ongoing involvement even when the child has moved out of the family home [48,51,59]. Two studies reported caregiving demands related to young adult vulnerability and safety concerns [48,53]. Lastly, one study discussed the stress related to health problems, especially when the child had a physical disability [48].

Regarding the caregivers’ objective burden factor, there was a mixed report on the effect on the caregivers’ employment; not all caregivers reported needing to be out of the workforce due to their caregiving needs [55,56]. When a comparison was made among caregiver needs of three different age groups (under 21, 21 to 30, and above 31 years old) of individuals with ID, there were significantly higher needs for caregivers caring for individuals with ID above 21 years old. Caregivers reported spending 15 out of 24 h on caregiving needs [55]. One study highlighted the financial burden and the need to manage practical household arrangements during the transition [57]. In summary, only one study indicated a significant difference between the caregiving hours before, during, and after the transition period, and few studies reported the specific factors contributing to the caregiving demands.

Regarding support and resources, two categories were reported, formal and informal. Regarding formal support, a consistent theme emerged in all 12 studies regarding the caregiver’s overall perceived dissatisfaction with adult, compared to child, services, even though some caregivers in each study reported being satisfied with individual service providers. The subthemes related to the dissatisfaction were a lack of information and lateness of the planning provided related to the transition (six studies), a lack of fit or accommodation to individual needs for adult services (five studies), caregivers’ needing to manage the transition on their own (three studies), a lack of adult support services compared to child services (three studies), a lack of parental involvement (two studies), a long waiting list for adult services (two studies), staff not adequately communicating (two studies), paperwork stress for the caregivers (two studies), possible incongruency between the professionals and caregivers’ understanding and expectations related to the transition (two studies), and one study each identified concerns such as a lack of “real” alternatives for vocational training providers, the application for accommodation service not being approved, a lack of coordination between child and adult services, professionals “not listening” to caregivers, negative experiences related to the environment and training in the employment service setting, requiring professional support for psychosexual development, the absence of respite care during the waiting period, service providers mistreating the caregivers or the child with ID, and the service workers being overworked with high turnover of staff.

Regarding informal support, there were mixed results on its effectiveness. Four studies reported that the caregivers benefited from informal support such as extended family members and friends, supportive networks, parent–child relationships, and parent groups. On the other hand, two studies reported that caregivers were reluctant to reach out and isolated themselves to avoid being burdensome to others and due to perceiving others to be fearful of their adult children, respectively.

Gauthier-Boudreault, et al. [52] labelled the caregivers’ needs for formal and informal support in four categories: material, informational, affective, and cognitive support. Material support is the need for resources to cater to practical needs during the transition, related to transition planning. Specific to the caregivers’ needs, caregivers expressed the need for informational support. It is necessary for information related to the transition, such as legal services, long-term planning, and service availability, for the caregivers to be more available and for aid with the decision-making process to be provided. Affective support would allow the caregivers to share experiences and socialise with other caregivers to cope emotionally. Lastly, caregivers reported the need for a proper fit of adult services to keep their children occupied and maximise their children’s ability through cognitive support. Due to the prolonged responsibility related to caregiving demands, all these support works may be needed concurrently for a “successful transition”, even after the transition period.

The coping mechanisms reported may be explicitly related to the transition period or overlap with formal and informal supports such as material, informational, affective, and cognitive support [52]. Only five of the twelve studies reported specifically on the coping mechanisms. Caregivers from three studies reported that they coped by gaining information related to the transition (informational support; [49,53,56]). Three studies highlighted the coping mechanism of, when needing to get access, taking a proactive role to reach for the support system, even though some caregivers reported hesitance (material support; [49,57,59]). One study reported that setting up a routine for the caregivers to volunteer in NGOs while the children attend the centres helped the caregivers to watch over their children and, at the same time, secure the children’s placement [57]. Lastly, another study reported that when caregivers had lower expectations, they could manage their worries better [53].

Based on the health approach, there has recently been more emphasis on the positive aspects of caregiving [28]. But only four of the twelve studies reported the positive aspects of caregiving factors. Two studies reported mixed feelings related to caregiving needs during the transition [48,57]. Not all caregivers reported only positive or negative experiences related to caregiving needs. The positive and negative experiences are possibly due to the fluctuation in the physical and social environmental factors and the intensity of the caregiving demands that the caregivers have to cope with [57]. The possible themes that had been highlighted were the sense of fulfilment and pride in their children and themselves [48,50]; gratitude towards at least one of the multiple service providers [49]; benefits for the caregivers (being more assertive and determined in meeting the transition needs enhanced caregivers’ resources and growth and ability to seek more formal and informal support [48]); and benefits for the individuals with ID (increased confidence and maturity, learning new coping skills, improvements in challenging behaviours, possible positive adjustment [48,49]).

Although possible positive aspects of caregiving were reported during the transition period, a perceived subjective burden was reported in eight of the eleven studies. Most of the studies reported negative emotions such as feeling physically and mentally tired of being the “fighter and advocate for their child” [48,50,51,58] and feeling helplessness or hopelessness, expressed in ways such as “feeling uncertain and confused” [48,49,57], being frustrated when “dealing with the inflexible and unresponsive service” [49], and having an “unsustainable burden of care” [56]. The caregivers reported that decision making was complex, and their daily activities felt restricted during the transition period. There was much uncertainty and a need to know what to expect [57]. Caregivers caring for children above 31 years old reported more negative feelings, evidenced in ways such as lower scores of well-being, more frustration, lower life satisfaction, and more sadness, compared to other caregiver groups caring for children who were below 30 years old [55]. Other reported feelings were guilt, resentment, anger, fear of repeat events and the future, depression, low self-worth, mistrust, and wishing that they do not have a child with a disability [48]. The caregiver could spend more time with their spouse only when the child left the family home [51]. The consistent themes linked to these negative feelings were the lack of support and dissatisfaction with the service system. There may have been a vicious cycle that made the caregivers feel physically and mentally tired, hopeless/helpless, and confused about what they could do during the transition period.

Some studies reported other factors that influence the caregivers’ quality of life during the transition. Four studies highlighted the caregivers’ concerns related to independence and vulnerability risk, such as the dilemma of letting go and separation [49,51,58,59]. Three studies also highlighted caregivers’ anxiety. Anxious caregivers tend to be overprotective in managing their high-need children, as the caregivers perceive that there is a lack of adequate support for the children [50,55,58].

In summary, the situation for caregivers of individuals with ID who are transitioning to adulthood is multi-faceted. One facet is that they have children who neuropathologically have potential ceiling effects on their cognitive abilities since birth and may also have multiple physical disabilities and health problems. Transitioning to adulthood adds more demands when there is a combination of biopsychosocial factors, possibly due to their physical development, psychological needs, and social and environmental settings. From the caregiver’s perspective, the caregivers are probably in the stage of life wherein they are more vulnerable to health and mental issues and more fragile than they used to be due to progressive ageing, interrupted career progression, possible retirement planning, financial needs, family dynamics, and other parenting needs. Possibly due to the incongruency between the services expected by caregivers (since there might generally be an increase in the care demands during the transition and/or based on caregivers’ expectation or understanding of “adulthood”) and the actual services that the system and professionals have offered, caregivers have consistently reported their overall dissatisfaction with adult services. Especially when the comparison with the child service system is made, the caregivers may feel that they are left alone, and that the responsibility for the child ultimately rests with them. Therefore, practical coping mechanisms such as information gaining, setting up a routine, taking a proactive role, accessing the support system, and managing their expectations are reported to help deal with the transition phase. There are also reported mixed feelings depending on the situation and the demands of care that the situation requires, even though most of the research studies emphasise the caregiver’s negative emotions and exhaustion in terms of being a caregiver compared to the positive aspects of caregiving. The vicious cycle of the increased demand during the transition, caregivers probably not being ready and hence unable to cope with the demands, concurrent with their perceived dissatisfaction with the services offered by the adult service system, leads the caregivers to develop helplessness. This cycle may be reinforced by the long waiting period to access the expected adult services, and caregivers continue to be overprotective and anxious due to the risk of vulnerability for their child.

### Transition Types

As Ferguson et al. [19] described, the transition to adulthood is a complex process with several key factors and groups of individuals involved. Based on this review’s findings, the expectations and contributions of the caregivers, the functional abilities and needs of the individuals with ID, and the formal and informal support systems, including professionals who work in the adult service centres, are expected to affect each other in this system. Thus, the bureaucratic transition, changes in professional and agency services, and the relationships formed (e.g., abandonment, surrender, assimilation, and engagement with professionals) with the caregiver during the transition may differ for each caregiver based on their circumstances. Those caregivers who gain sufficient information and can take a proactive role during the transition [57,62,63] may benefit more when they collaborate with the professionals (engagement with professionals) or provide needed services for their child (assimilation with professionals). On the other hand, those caregivers who experience a high burden of care due to other stressors may not be able to meet the commitment needed for the transition and, hence, may not benefit from the relationship with the professionals [15,64].

The outcome of the bureaucratic transition and the perceived relationship with professionals may spill over into the family life transition. Based on the adult service placement outcome, the caregivers and the family must accommodate their routines and roles within the family. The caregivers and other family members are expected to cope with the changes and make any alternative arrangements needed in their family home setting, if the adult service placement is inaccessible or unsuitable. It is a possible theme, as reported by the caregivers, that they were not prepared for the transition and felt hopeless when the cycle was reinforced by the prolonged waiting period [65].

Finally, the status transition is the status change from child to adult. This review’s findings on the caregiver’s apprehensiveness related to independence and risk of vulnerability align with Ferguson et al.’s [19] study that the status transition could be more related to how much more or less supervision their child with ID needs, rather than them expecting that the child will be able to gain total independence as they reach the age of 21. Some caregivers viewed the individual with ID as a “little child,” not a teenager. Hence, it may be more helpful for the caregivers to view the transition from a more constructive perspective, which includes all stakeholders and focuses on deciding the “right thing to do”, rather than categorising what is “right” and “wrong” concerning what an adult would do [59,66].

## 8. Limitations and Future Directions

The studies included in this scoping review focused on the caregiver’s adaptation to adult services when their child is known by the service providers. Even though the literature on “transition to adulthood” was considered with the intention to broaden the search and capture literature on individuals with ID who did not attend school or adult services, the findings from this scoping review may only generalise to caregivers caring for individuals with ID who attended special education, completed their schooling, and transitioned into adult services. The data from the caregivers of individuals who dropped out of the school system and are unknown to the adult service system have not been captured. These cases may include an extreme spectrum of caregivers coping with the transition and not approaching the adult services for extra support or who have not been coping and did not want to approach the adult services. Further, caregivers caring for their children with challenging behaviours since childhood may have a different profile of needs. These caregivers would have sought support earlier in their children’s lives, such as children’s group homes and respite care. Therefore, future studies may consider recruiting caregivers from various sources, such as child guidance and development assessment centres, hospitals, and respite care and community-based service settings such as group homes, as this review did not find any studies representing this population of children and their caregivers.

There was only one quantitative study and eleven qualitative exploration studies. Most of the studies were designed based on the explorative qualitative method to gain a broad and deeper understanding of the caregivers’ needs during the transition via open-ended questions. The qualitative research method is valuable in this under-investigated area of caregivers’ well-being, specifically during the transition to adulthood, as it will help identify key factors that can be used as a base for more critical analysis via quantitative research methods [67]. Quantitative studies allow for comparing the caregiver groups’ needs to understand the significant differences between the critical factors in this area [67]. Therefore, future study research design should involve quantitative methods to assess for significant differences between caregivers providing care for individuals with ID across the three stages of pre-transition, during transition, and post-transition to adulthood. Additional research could compare these three stages with caregivers caring for young adults with typical development.

Regarding this scoping review methodology, only literature published in English was included. Future review methodology may include non-English literature for more comprehensive review findings.

## 9. Conclusions

Although most of the studies reviewed were based on qualitative findings, the explorative nature of the study design did capture as broadly as possible the factors that affect caregivers’ adaptation process in caring for individuals with ID during the transition to adulthood. Although there were no studies on the significance of the increase in caregiving demands, specifically during the transition to adulthood, one study highlighted the significant need to spend more hours on caregiving during the transition period. However, the theme regarding caregivers’ dissatisfaction with formal adult services was consistent and, in a vicious cycle, affected caregivers’ negative emotions related to caregiving needs, increasing their dissatisfaction. Some studies highlighted the “mixed feelings”, including the positive factors related to the caregiving experience, depending on the setting factors. This review’s findings highlighted caregivers’ apprehensiveness relating to the vulnerability risk and definition of independence for their children. The coping mechanisms discussed in this review need to be studied in more detail but provide direction for a practical approach to meeting the demands of this particular caregiving role. The findings in this review may suggest the importance of preparing and educating caregivers before their children graduate from child services about the possible changes in caregiving demands, increased vulnerability risk, and ways of dealing with adult services during the transition period. This may aid the caregivers in forming their definitions of independence, with possible shifts in expectations and comparison with the typical definition of adulthood. Future research may focus on quantitative studies comparing caregivers providing care for individuals with ID across the three stages of pre-transition, during transition, and post-transition to adulthood. Additional research could compare these three stages with caregivers caring for young adults with typical development. There is a need for more information to understand the relationship between caregiving demands and caregivers’ perceptions or sense of helplessness that significantly affect their quality of life.

## Figures and Tables

**Figure 1 healthcare-12-00116-f001:**
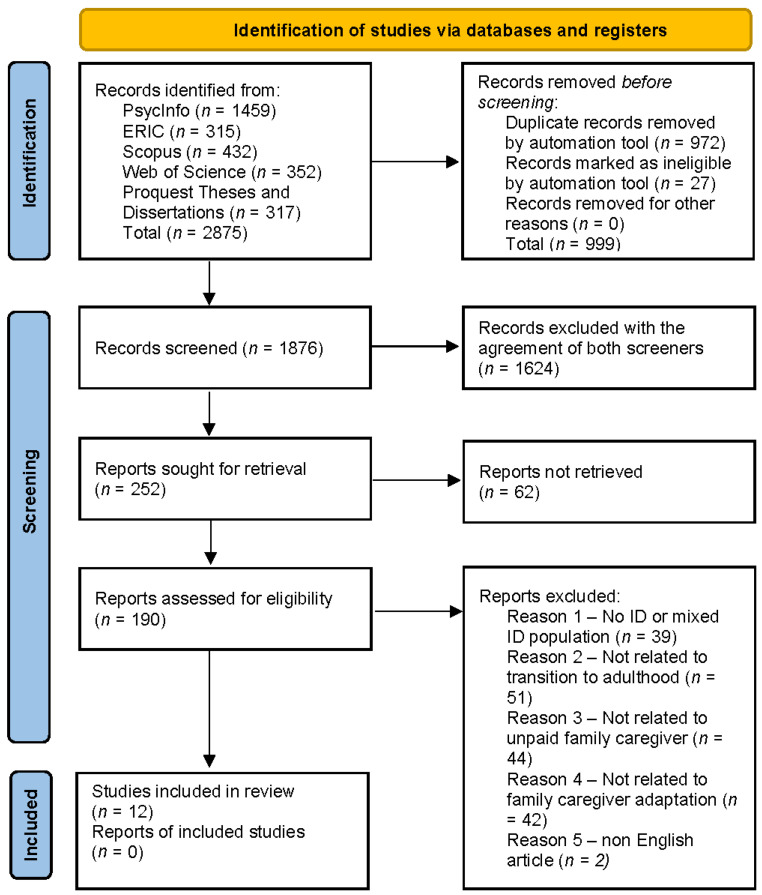
Outline of screening process according to PRISMA 2020 flow chart. Adapted from PRISMA 2020 flow chart.

**Table 1 healthcare-12-00116-t001:** Characteristics of the studies.

Author(s)/(Year of Publication)	Country	Aim(s)	Type of Study	Individual with ID (1) Diagnosis—Range (2) Age of the Individual with ID (3) Comorbid Diagnoses	Caregivers (1) Sampling (2) Number of Caregivers (3) Age Range of Caregivers (4) Ethnicity
Rapanaro, C. et al., (2008) [48]	Western Australia	To investigate the perceived benefits and negative impacts associated with stressful events and chronic caregiving demands encountered by parents caring for young adults with an intellectual disability in the period of transition to adulthood	Qualitative content analysis	(1) ID Mild, moderate, and severe or profound range (2) 16 to 21 years old (3) Not stated	(1) Purposive sampling (2) 119 parents (3) Mean age: 48.05 years old (4) Not stated
Gillan, D. et al., (2010) [49]	Ireland	To gain understanding of the impact on parents of the transition from high-support school environments to mainstream settings with potentially lower levels of formal supports	Qualitative Grounded theory (explorative in nature) Cross-sectional Semi-structured interview	(1) ID Mild range (2) 19 to 24 years old (3) Not stated	(1) Purposive sampling (2) 12 parents; 4 married couples and 4 single mothers (3) 42 to 65 years old (4) Not stated
Hubert, J. (2011) [50]	England	To gain understanding of the experiences and perspectives of families, especially mothers, of young people with these complex needs, including attitudes towards long-term residential care	Qualitative Ethnographic study Informal interview and participant observation—over a period of 2 years (longitudinal)	(1) ID Severe to profound range (2) 15–22 years old (3) Three-quarters had epileptic episode	(1) Not stated (2) 20 parents (3) Not stated (4) Not stated
Isaacson, N. C. et al., (2014) [51]	Australia	To gain understanding of the future the young people and their families were seeking, important issues faced, difficulties and supportive factors, the impact on family relationships, and family perception of the purpose of the Community Living Plan (CLP) during this transition	Qualitative 2 case studies Interviews, observation and documentation review over a period of 7 months (longitudinal study)	(1) Down syndrome—moderate to high support needs (2) 21 and 25 years old (3) Not stated	(1) Purposive sampling (2) Both father and mother involved (3) Not stated (4) Not stated
Gauthier-Boudreault, C. et al., (2017) [52]	Canada	To document the needs of parents and young adults with profound ID during and after the transition to adulthood by exploring their transitioning experience and factors that influenced it	Qualitative Descriptive–interpretative approach	(1) ID Profound range (2) 18 to 26 years old (6 young adults still in school and 8 post-school) (3) Not stated	(1) Purposive and snowballing sampling (2) 14 caregivers; 12 mothers and 2 fathers (3) 49 years old and below: 5 caregivers 50 years old and above: 9 caregivers (4) Not stated
Biswas, S. et al., (2017) [53]	United Kingdom	To explore parents’ retrospective views of their child’s developmental transition into adulthood, and how parents adjust and adapt to this transition	Qualitative Retrospective cross-sectional exploratory design	(1) ID Severe range (2) 19 to 57 years old (3) Physical and sensory disability; some had physical care needs	(1) Non-probabilistic purposive sampling (2) 12 parents of 11 children; 7 mothers, 3 fathers, 1 stepmother, 1 stepfather (3) 44 to 78 years old (4) White British
Wilcox, G. et al., (2019) [54]	Canada	To understand the particular experiences of two mothers and their perspectives on the process of transitioning from high school to adulthood for their children with intellectual disability (ID)	Qualitative Exploratory; 2 case studies	Case study 1: Male, mild ID, attending postsecondary education program, living independently in postsecondary residence Case study 2: Female, severe ID, supported group home	(1) Purposive sampling (2) 2 mothers (3) Not stated (4) Not stated
Gur, A. et al., (2020) [55]	Israel	To fill an important gap in the caregivers’ well-being literature by focusing specifically on families of children with ID who are navigating the transition to adulthood	Quantitative; individual interview	(1) ID Mild to profound range (2) 3 age groups: Group 1: Under 21 years old Group 2: 21–30 years old Group 3: 31 years old and above (3) Not stated	(1) Purposive sampling (2) 301 participants; 256 women and 41 men (3) Mean age: Group 1: 42.24 years old Group 2: 54.48 years old Group 3: 63.07 years old (4) Israeli
Roos, E. et al., (2020) [56]	Norway	To identify factors that improve the collaboration process between parents and employees, creating less burden for the parents of child with profound ID	Descriptive qualitative study Semi-structured interview; face-to-face individual interviews, or group interviews	(1) ID Profound range (2) 3 age groups: 18–20, 20–25, >25 years old (3) Not stated	(1) Purposive sampling (2) 9 parents; 7 mothers and 2 fathers (3) Not stated (4) Not stated
Ellman, E. et al., (2020) [57]	South Africa	To describe how parents experienced the transition from special school to post-school of their children with severe intellectual disability in a small town in the Western Cape	Qualitative; 5 case studies	(1) ID Severe range (2) 18–35 years old (3) Severe ID and some comorbid with Down syndrome or cerebral palsy	(1) Purposive sampling (2) 5 parents; 3 mothers and 2 fathers (3) Not stated (4) Not stated
Codd, J. et al., (2021) [58]	United Kingdom	To explore parental experiences of having a son/daughter with an intellectual disability transition to adulthood and what meaning parents make of this	Qualitative; semi-structured interview Cross-sectional Interpretive phenomenological analysis Learning Disability Screening Questionnaire (LDSQ) to validate diagnosis of ID	(1) ID Mild to profound range (2) 18–23 years old (3) Down syndrome, ASD, ADHD, Williams syndrome, FOXG1 syndrome	(1) Purposive sampling (2) 10 parents; 7 mothers and 3 fathers (3) 40 to 65 years old (4) British and Italian
Nucifora, A. et al., (2022) [59]	Australia	To examine parents’ perceptions of adulthood for their children with an ID, as well as their experience of the child’s transition to adulthood	Descriptive–interpretive qualitative; semi-structured interview Thematic analysis	(1) ID No severity indicated (2) 17–42 years old (3) Autism, Prader–Willi syndrome, Down syndrome, psychosis, genetic disorder	(1) Purpose and snowball sampling (2) 8 parents; 5 mothers, 2 fathers, and 1 kinship carer (3) 50–75 years old (4) Not stated

## Data Availability

Not applicable.

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
