# Peer review of "Family Caregiver Adaptation during the Transition to Adulthood of Individuals with Intellectual Disabilities: A Scoping Review"

_healthcare, 2024, doi:10.3390/healthcare12010116_

Round 1

Reviewer 1 Report

Comments and Suggestions for Authors

I read the manuscript by Shivasangarey Kanthasamy et al. entitled "Family Caregiver Adaptation during the Transition to Adulthood of an Individual with Intellectual Disability: a Scoping Review".   The paper offers a scoping review that examines the breadth of available research literature on family caregiver adaptation in the care of people with ID transitioning to adulthood of the 2875 articles identified, 12 published studies were included in the final review. Overall, the authors concluded that family caregivers experienced increased demands during the transition to adulthood of a person with ID who was not prepared to cope with such increased demands, while at the same time being dissatisfied with the adult service system, leading caregivers to develop a pervasive sense of helplessness.

This is an interesting and thorough review of an ever-timely topic. The researchers used an appropriate methodology to explore the questions posed in the introduction. I believe the paper can be published after minor revision.

My comments follow:

-While there are, labeled, Methods and Results sections in the manuscript, the labeling of the initial section as Introduction is missing and the same is true for the Discussion section (the title in 6.3. "Summary" makes no sense, please replace it with "Discussion").   -In lines 34-35, why is the title repeated? Please replace it with "Introduction".

-What was the time frame, which included the literature review, did you research the publications from when to when? (No suitable studies were found before 2008 or you did not search for older publications).

 - Table 2 needs to be improved, consider splitting it into more tables to make it easier for the reader to read.

-Section 6 should be shortened. Please provide a shorter description of the studies.

- Customizing references according to MDPI guidelines is something the journal will also ask you to do.

Reviewer 2 Report

Comments and Suggestions for Authors

I would like to thank the editors and authors for the opportunity to review the article "Family Caregiver Adaptation during the Transition to Adulthood of an Individual with Intellectual Disability: A Scoping Review"

In general terms, I can say that the article presents specific references in interest, however most of them in the introduction and justification of the study are more than 5 years old  (more than 73%).

The chosen theme is interesting and of greater importance.

I will now offer my contributions or suggestions for improving the manuscript:

Lines 34-35

In the point "1. Family Caregiver Adaptation During the Transition to Adulthood of an Individual with Intellectual Disability: A Scoping Review, I suggest removing ": A Scoping Review" leaving just "1. Family Caregiver Adaptation During the Transition to Adulthood of an Individual with Intellectual Disability"

Lines 34-116

In the introductory part where concepts are explored, review and introduce more recent references, preferably less than 6 years old.

Linha 241

The authors refer to "Literature published only in English", this aspect should be considered a limitation of the present study and should appear in the limitations chapter.

Lines 339-343; 355-356; 402-404; 490-492; 502-507; 540-541; 571-572; 659-663; 685-688; 709-712; 727-728

The authors insert direct quotations into the manuscript, and do not include the page. And in some the authors and year are also missing.

6. Results; 6.2.1. Objective 1, … Objective 2,3,4…

In the results chapter, the authors subdivide these by objectives, however it seems to me that they divide them by research questions, as they only present a single objective " The objective of this scoping review 163 was to explore the breadth or extent of the available research literature and to map and 164 summarise the evidence from the literature on family caregiver adaptation, specifically 165 during the transition to adulthood of an individual with ID." (lines 163-166).

7. Limitations (lines 929-958)

The authors discuss the limitations of the studies found. However, they need to discuss the limitations of the scoping review process, for example, they opted for articles only in English.

Final decision:

The manuscript needs minor changes.

I hope that my contributions will serve to improve this article and the study you propose.

Reviewer 3 Report

Comments and Suggestions for Authors

Thank you very much for give me the opportunity to review this interesting paper titled Family Caregiver Adaptation during the Transition to Adulthood of an Individual with Intellectual Disability: A Scoping Review. The paper offers a scoping review studies the breadth of the available research literature on family caregiver adaptation in caring for individuals with ID transitioning to adulthood and was conducted according to Joanna Briggs Institute (JBI) Scoping Review recommended methodology guidelines. Of 2875 articles identified, 12 published studies were included in the final review. In overall authors concluded that Family caregivers experienced increased demands during the transition to adulthood of an individual with intellectual disability not being prepared to cope with such increased demands, while concurrently they were dissatisfied with the adult service system, leading the caregivers to develop a pervasive sense of helplessness.

Altogether it is an interesting study quite well written. The review carried out by the authors is quite exhaustive and their work contains a lot of information, and in my humble opinion it is a paper that could contribute to the field. However, I think it would be necessary and interesting for the authors to reorient and rewrite it various aspects offering greater clarity and structure in their discourse making it more readable. Say this, I have some concerns that I detail bellow.

1.     Perhaps it could be useful if you review the wording of the abstract, placing the question addressed in a broad context and highlighting more clearly the purpose of the study. It contains more than 200 words, the maximum indicated by the journal.

2.     It does not appear that the authors follow the structure recommended by the journal: on lines 34-35 the title of the paper appears again, and not what it might usefully contain: a introduction that briefly placed the study in a broad context and highlights why it is important.

3.     In a similar vein, I think it could be interesting to strengthen the paragraphs 2 and 3 by helping the reader, so that they follow and value their work more, for example, explaining a little bit more the pathogenic paradigm of illness-focused behaviour and the health paradigm.

4.     I'm a bit confused about the use of labels: in section four you talk about research questions, in 6.2 about objectives.

5.     In section 5.1.1., we would be talking about participants in the reviewed studies as an inclusion criterion?

6.     In general, in section 5 could be of interest a little more help for the reader to follow him, as it is a complex section with several sub-headings.

7.     In a similar vein the location of table 2 is not advanced to the reader, it would also be interesting to review its format and length.

8.     In point 6 the authors offer a detailed analysis of the studies reviewed. However, in my humble opinion, it would be interesting to have a less descriptive analysis with a greater synthesis and contribution, which could also be useful to reduce the length of the work and make it easier to read.

9.     In my humble opinion, it would be interesting to include a discussion paragraph. Perhaps you could reorient subheading 6.3, starting by recalling the objective of your work, discussing with the authors, and adding some more arguments about the practical implications and suggestions for future research of their work. In your limitations section you seem to highlight mainly those of the studies you have reviewed, could you elaborate on what yours would be and what suggestions you derive from them for future research?

10.  Maybe the conclusion section might be improved, in example, adding some more arguments regarding the practical implications and suggestions for future research of your work.

11.  Please check the format (for example lines 249-250, 260, it seems there are extra parentheses or spaces)

12.  Please review the references, both in the text and in the references section, as they do not seem at all to follow the format indicated by the journal. Also, if possible, it would be interesting to incorporate a greater number of references, especially more recent ones (only three dates from 2022 and none from 2023).

Round 2

Reviewer 3 Report

Comments and Suggestions for Authors

Thank you very much for give me the opportunity to review again this interesting paper titled Family Caregiver Adaptation during the Transition to Adulthood of an Individual with Intellectual Disability: A Scoping Review. The paper offers a scoping review studies the breadth of the available research literature on family caregiver adaptation in caring for individuals with ID transitioning to adulthood and was conducted according to Joanna Briggs Institute (JBI) Scoping Review recommended methodology guidelines. Of 2875 articles identified, 12 published studies were included in the final review. In overall authors concluded that Family caregivers experienced increased demands during the transition to adulthood of an individual with intellectual disability not being prepared to cope with such increased demands, while concurrently they were dissatisfied with the adult service system, leading the caregivers to develop a pervasive sense of helplessness. In my impression the authors have improved their work considerably, although I still find the number of recent citations that have been included to be too few. 

Author Response

Following this suggestion previously, we conduct a search using the Web of Science database for recent (last five years) citations of the papers already contained in the Introduction. We found five recent papers that were suitable and have included them in the Introduction of the revised manuscript.

If the reviewer has specific papers they would like to nominate then we are happy to consider them. But we feel that the search strategy we have used to find recent, relevant papers is a valid one and we have already included the relevant findings from this in our manuscript.